# PROPER ORTHOGONAL DECOMPOSITION FOR SCALABLE TRAINING OF GRAPH NEURAL NETWORKS

## ABSTRACT

As large-scale graphs become ubiquitous in real-world applications, there is growing concern about the memory and time requirement to train a graph neural network (GNN) model for such datasets. Storing the entire adjacency and node embedding matrices in memory is infeasible in such a scenario. Standard sampling-based methods for addressing the memory constraint suffer from the dependence of the number of mini-batches on the graph size. Existing sketch-based methods and graph compression techniques operate at higher sketch ratios, with the graph compression techniques showing poor generalization, implying that different GNNs trained on the same synthetic graph have performance gaps. Sketch-based methods necessitate online learning of sketches, further increasing the complexity. In this paper, we propose a new sketch-based algorithm, PGNN, employing the Proper orthogonal decomposition (POD) method to craft update rules to train GNNs, improving the memory requirement and training time without the complication of updating the sketches during training. Experiments on standard graph datasets show that PGNN can reach much lower sketch ratios without compromising the performance. We prove the optimality of the POD update rule for the linearized GNN (SGC). Empirical findings validate our approach, demonstrating superior performance at reduced sketch ratios and adaptability across various GNN architectures.

## 1 INTRODUCTION

Graph Neural Networks (GNNs) have proven to be powerful tools for graph learning across various domains, excelling in tasks such as classification Kipf & Welling (2017), clustering Bianchi et al. (2020), recommendation systems Wu et al. (2022), and social network analysis Fan et al. (2019). Their strength lies in their ability to extract meaningful insights from local neighbourhoods within graphs, thus creating effective representations of target nodes. However, the dependence of GNNs on graph topology introduces significant challenges when scaling to larger graphs or deeper models while maintaining computational and memory efficiency. Traditional full-batch training methods necessitate storing the Laplacian matrix of the entire graph, resulting in a memory complexity of $O(m + ndL + d^2L)$ for an $n$-node, $m$-edge graph, where node features have dimension $d$ in an $L$-layer *graph convolutional network* (GCN). This linear dependency on both $n$ and $m$, combined with the limited memory capacity of GPUs, restricts the scalability of training on large graphs (especially large dense graphs with $m$ being of the order of $O(n^2)$ in the worst case). To address these memory constraints, research in this domain has broadly proposed two main approaches: sampling-based strategies Hamilton et al. (2018); Chen et al. (2018a;b); Chiang et al. (2019); Zeng et al. (2020) and historical embedding techniques Fey et al. (2021); Ding et al. (2021). Although these methods improve memory efficiency, the computational complexity still increases linearly with $n$ and $m$.

In the context of matrix approximation, a *sketch* of an arbitrary matrix $A \in \mathbb{R}^{n \times d}$ is defined as a much reduced matrix $B \in \mathbb{R}^{c_0 \times d}$, where $c_0$ denotes the *sketch dimension* which is significantly smaller than $A$ (i.e., $c_0 \ll n$) but still provides a good approximation Ghashami et al. (2015). Here, $c_0$ denotes the *sketch dimension*. The amount of compression achieved by sketching is best described by the *sketch ratio* $r = c_0/n$. *Proper orthogonal decomposition* (POD) Rathinam & Petzold (2003),

also known as the Karhunen–Loéve decomposition or principal component analysis, provides an orthonormal basis representing the given data in an optimal least squares sense.

To achieve sublinear training time complexity with respect to $n$, Ding *et al.* Ding et al. (2022) propose a sketch-based algorithm named *sketch-GNN*, that trains the GNN on top of a few compact sketches of both the *convolution* and *node feature* matrices. The authors propose an end-to-end training protocol in the sketch space by approximating the non-linear activation function using *polynomial tensor-sketch* (PTS) theory Pham & Pagh (2013).

As observed by the authors, the approximation of the non-linear activation limits the expressiveness, which constrains the depth of GNNs that can be trained due to error accumulation. Working with dense matrices, even in the reduced sketch space, imposes a significant computational burden. Despite showing promise, the sketch-GNN algorithm needs to: (i) work at a *higher sketch ratio* which results in a lower compression of the original graph and (ii) *requires frequent updates* of the sketches during the training which triggers re-computation of all the sketches involved. It is worthwhile to mention another technique which guarantees efficient training of GNNs using random spanning trees Bonchi et al. (2024) and leverages the concept of effective resistance to enhance node classification tasks. This method enhances GNN efficiency by creating path graphs from random spanning trees to maintain essential graph features while minimizing complexity for faster training. This approach is currently constrained to only the GCN architecture.

The primary motivation for using POD to address limitation (i) in sketch-GNN is that, in dynamical systems, the effective number of eigenmodes required decreases as the system size increases, which results in a lower sketch ratio (i.e., the sketch sizes are much smaller). Relevant works, such as Choi et al. (2023), demonstrate a clear connection between message-passing methods and dynamical systems. For limitation (ii), we theoretically establish bounds that constrain the deviations of node representations when using the PGNN method. Additionally, we prove the optimality of the POD update rule for the linearized GNN update rule, indicating that the best low-rank matrix for the update rules can be predetermined, which completely eliminates the need for online learning via frequent updates of sketches. To this end, our paper presents **PGNN**, a novel sketch-based method for GNNs. This method diverges fundamentally from prior approaches that emphasize sketching weights or gradients (see Liu et al. (2022), Chen et al. (2015), Kasiviswanathan et al. (2018), Lin et al. (2019), Spring et al. (2019)). Drawing inspiration from the update rules of linearized GNNs (SGC) Wu et al. (2019), we customize the message passing process to function within the linear subspace formed by the columns of the augmented input node feature matrix. Experimental results, as presented in section 5, demonstrate that the sketch ratio necessary for achieving optimal performance decreases as the graph size increases. Despite its theoretical optimality limitations beyond linearized GNN, the PGNN framework efficiently performs node classification in POD-derived linear subspaces, providing insights into GNN operational subspaces Lee et al. (2023).

Our contributions can be summarized as follows:

1. In section 3, we introduce specialized update rules designed to enhance the training efficiency of GNNs by operating within a reduced subspace. Utilizing the POD method, we sketch the input node feature and convolution matrices into their lower-dimensional approximations, thereby streamlining the computational process.

2. In Theorem 1, we establish the optimality of the POD method in the linearized update rule of the GNN. Furthermore, in Lemma 2, we present bounds that quantify the deviation of node representations when using the PGNN framework.

3. The versatility of PGNN is evaluated across different GNN architectures, including GCN Kipf & Welling (2017), SGC Wu et al. (2019), GraphSAGE Hamilton et al. (2018), and GAT Veličković et al. (2018), with results detailed in section 5. Through extensive experimentation, as demonstrated in section 5, we find that PGNN is able to work with reduced sketch ratios. For instance, on the Reddit dataset, the state-of-the-art sketch-GNN framework achieves its highest accuracy at a sketch ratio of 0.3, while we achieve a comparable accuracy for a much lower sketch ratio of 0.05, which in turn results in a much faster algorithm with lower memory requirement.

## 2 PRELIMINARIES

### BASIC NOTATIONS.

Let $G = (V,E)$ denote a graph where $V = [n] := \{1, \ldots, n\}$ is the set of $n$ vertices and $E \subset V \times V$ is the set of $m$ edges. Additionally, the input node feature matrix associated with $G$ is denoted by $X^{(0)} \in \mathbb{R}^{n \times d}$, where $d$ is the number of features. Let $\tilde{X}^{(0)} \in \mathbb{R}^{n \times c_0}$ and $\bar{x}$ denote the *augmented feature matrix* and its *mean vector*, respectively. $C \in \mathbb{R}^{n \times n}$ denotes the convolution matrix of graph $G$ and $C(i,j)$ denotes its $(i,j)$-th entry. We represent the $k^{th}$ order element-wise power of $C$ as $C^{\odot k}$. Additionally, $C(i,:)$ denotes the $i^{th}$ row and $C(:,j)$ denotes the $j^{th}$ column.

Considering a GNN, $X^{(l)} \in \mathbb{R}^{n \times d_l}$ denotes the node representations of layer $l$, where $d_l$ represents the number of neurons at layer $l$. $\|\cdot\|$ denotes the Frobenius norm unless stated otherwise. $\sigma(\cdot)$ is the non-linear activation and $\Theta^{(l,q)}$ is the learnable weight matrix at layer $l$ for filter $q$.

$R^{(1)}, R^{(2)}, \ldots, R^{(k)}$ denote the $k$ count-sketch matrices with dimension $\mathbb{R}^{c_k \times n}$, where $k$ (a hyperparameter) denotes the number of sketches and $c_k$ is the fixed sketch dimension associated with all of them. $\beta$ denotes the upper bound on the number of elements in the set for unsketching.

The POD projection matrix which is the matrix of the linear projection expressed in the original coordinate system in $\mathbb{R}^n$ is given by $P = \rho^T \rho \in \mathbb{R}^{n \times n}$. We refer to the submatrix $\rho$ as the *factor* of $P$. We represent a matrix comprising of $b$ $n$-dimensional column vectors $y \in \mathbb{R}^n$ as $[y]_{n \times b}$.

### COUNT SKETCH.

Matrix multiplication is crucial in machine learning and scientific computation, with efficient techniques developed in works like Paszke et al. (2017), Guennebaud et al. (2010), and Abadi et al. (2016). Count sketch, a potent dimensionality reduction technique introduced in Charikar et al. (2002) and Weinberger et al. (2010), projects an $n$-dimensional vector $u$ into a $c_k$-dimensional space using a random hash function $h : [n] \to [c_k]$ and a binary Rademacher variable $s : [n] \to \{-1, 1\}$. The dimension reduction transformation $CS(u)_i = \sum_{h(j)=i} s(j)u_j = R(i,:)u$ involves a count sketch matrix $R \in \mathbb{R}^{c_k \times n}$.

### LOCALITY SENSITIVE HASHING.

Locality Sensitive Hashing (LSH) exploits hash functions, denoted as $H : \mathbb{R}^d \to [c_0]$, to map closely positioned vectors into the same bucket with high probability. SimHash, an instance of LSH, uses a random matrix $P \in \mathbb{R}^{c_0/2 \times d}$ to define a hash function $H(u) = \arg\max([Pu \,\|\, Pu])$ Charikar et al. (2002). This method is efficient for large vector batches Andoni et al. (2015). We use the LSH technique Chen et al. (2020) without online updates to the hash matrix $P$.

### PROPER ORTHOGONAL DECOMPOSITION.

Given the input node feature matrix $X^{(0)} = [x_1, x_2, \ldots, x_d]$, where $x_i \in \mathbb{R}^n$. Then the best approximating affine subspace representing these data points and passing through the mean ($\bar{x} = \frac{1}{d} \sum_{i=1}^d x_i$) is given by the leading eigenvectors of the *centred covariance matrix* (see Rathinam & Petzold (2003) for a detailed explanation)

$$\bar{R} = \frac{1}{d-1} \sum_{i=1}^d (x_i - \bar{x})(x_i - \bar{x})^T.$$

The factor $\rho \in \mathbb{R}^{c_0 \times n}$ of projection $P$ is given by the leading eigenvectors of $\bar{R}$, where $c_0 \ll n$. The sketch $Z^{(0)} = \rho(X^{(0)} - [\bar{x}]_{n \times d}) \in \mathbb{R}^{c \times d}$ of input node feature matrix $X^{(0)}$ represents the sketch of $X^{(0)}$ in the affine subspace. To know more about POD, refer to (Rathinam & Petzold (2003), Holmes et al. (1996), Lall et al. (1999), Moore (1981)).

UNIFIED FRAMEWORK OF GNNs.

For a GNN, Message passing between layers can happen differently, like that of spatial convolution (GCN)Kipf & Welling (2017), self-attention (GAT)Veličković et al. (2018), and Weisfeiler-Lehman (WL) alignment, see Xu et al. (2019). The general rule according to Balcilar et al. (2021) for message passing is given by,

$$X^{(l+1)} = \sigma \left( \sum_q C^{(l,q)} X^{(l)} \Theta^{(l,q)} \right), \tag{1}$$

where $C^{(l,q)} \in \mathbb{R}^{n \times n}$ is the $q$-th convolution support at layer $l$ that defines how the node features are propagated to the neighbouring nodes, $X^{(l)}$ is the node representations at layer $l$, and $\Theta^{(l,q)}$ are the trainable weights. The input node feature matrix is given by $X^{(0)} \in \mathbb{R}^{n \times d}$.

The gradient involved in the back-propagation rule for GNNs, as shown in Ding et al. (2021) for the loss function $\ell$, is given by the following:

$$\nabla_{X^{(l)}} \ell = \sum_q \left( C^{(l,q)} \right)^\top \left( \nabla_{X^{(l+1)}} \ell \odot M^{(l+1)} \right) \left( \Theta^{(l,q)} \right)^\top. \tag{2}$$

$M^{(l+1)} = \sigma' \left( \sigma^{-1} \left( X^{(l+1)} \right) \right)$. This formulation embodies the essence of the message-passing paradigm. Here, $\sigma'$ and $\sigma^{-1}$ denote the derivative and the inverse of the activation function $\sigma$, respectively. The term $\nabla_{X^{(l+1)}} \ell \odot \sigma' \left( \sigma^{-1} \left( X^{(l+1)} \right) \right)$ represents the gradients propagated back through the non-linearity. In essence, this rule captures the flow of information and updates dynamics within GNNs during the backward pass.

## 3 POD SKETCH BASED METHOD ON GNNs

**Problem and Insights.** The runtime complexity of the update rules of GNNs on a complete graph is $O(n^2)$, and the memory complexity involved is $O(n + m)$. The POD sketch-based method for GNNs approximates the GNN's update rule and utilizes sketches of both the convolution matrix and the input node feature matrix for training. Initially, the input node feature matrix ($X^{(0)}$) and the convolution matrix ($C = \tilde{D}^{-1/2} \tilde{A} \tilde{D}^{-1/2}$) are of sizes $n \times d$ and $n \times n$, respectively. These matrices are then transformed into low-dimensional sketches of size $c_0 \times d$ and $c_0 \times c_0$, respectively. The sketch $Z^{(0)}$ of the input node feature matrix $X^{(0)}$ and the *convolution matrix sketch* ($S_C$) are given by:

$$Z^{(0)} = \rho(X^{(0)} - [\bar{x}]_{n \times d}), \quad S_C = \rho C \rho^T.$$

Recall that $\rho$ is the *factor* of $P$, representing the singular vectors of the augmented input node feature matrix $\tilde{X}^{(0)}$ after normalization (see Algorithm 1).

### 3.1 APPROXIMATE UPDATE RULES WITH PGNN

Our primary goal is to sketch the forward propagation

$$X^{(l+1)} = \sigma \left( C X^{(l)} \Theta^{(l)} \right).$$

We project the node representations at layer $l$ onto the subspace spanned by the columns of the factor matrix $\rho$, giving us

$$\tilde{Z}^{(l+1)} = \rho \sigma \left( C \left( \underbrace{\rho^T Z^{(l)} + [\bar{x}]_{n \times d_{(l+1)}}}_{\text{T}} \right) \Theta^{(l)} \right) - \underbrace{[\rho \bar{x}]_{c_0 \times d_{(l+1)}}}_{\text{U}}$$

We denote the bias factor induced by this projection as $U = [\rho \bar{x}]_{c_0 \times d_{(l+1)}}$. The dependence of the convolution matrix on $n$ is not removed; hence, we project and inverse project for terms inside the non-linear activation to obtain equation 3. For ease of notation, let

$$T = [\bar{x}]_{n \times d_{l+1}}, \quad W^{(l)} = \left( S_C Z^{(l)} + M \right), \quad M = [\rho C \bar{x}]_{n \times d_l}.$$

$$Z^{(l+1)} = \rho\sigma\left(\rho^T\left\{W^{(l)}\Theta^{(l)} - U\right\} + T\right) - U \tag{3}$$

The general update rule for the PGNN framework is:

$$Z^{(l+1)} = \rho\sigma\left(\rho^T\left(\sum_q \left\{W^{(l,q)}\Theta^{(l,q)} - U\right\}\right) + T\right) - U \tag{4}$$

where $W^{(l,q)} = \left(S_C^{(l,q)}Z^{(l)} + M^{(l,q)}\right)\Theta^{(l,q)}$ and $M^{(l,q)} = [\rho C^{(l,q)}\bar{x}]_{c_0 \times d}$. The mean of the augmented input feature matrix $\tilde{X}^{(0)}$ (See Algorithm 1) is denoted by $\bar{x}$. The sketch of the $q$-th convolution matrix at layer $l$ is given by $S_{C^{(l,q)}} = \rho C^{(l,q)}\rho^T$, and the node representations at layer $l$ given by the PGNN method is denoted by $Z^{(l)} \in \mathbb{R}^{c_0 \times d_l}$. The weight matrix at layer $l$, filter $q$ is denoted by $\Theta^{(l,q)} \in \mathbb{R}^{d_l \times d_{l+1}}$, where $d_l, d_{l+1}$ denotes the hidden layer dimensions at layers $l, l+1$. The intricacies of how message passing happens for the PGNN framework in various GNN architectures like SGC, GCN, GraphSAGE, and GAT are explained in Appendix B. Two challenges must be addressed for the approximate update rule proposed equation 4.

> **Challenge (1).** Dependence of matrix $\rho$ on the number of vertices $n$ is a bottleneck.

**Addressing challenge (1)** The rise in popularity of *approximate matrix multiplication* (AMM) stems from its adaptability to large-scale datasets, rendering matrix computation more feasible. The count-sketch method discussed in section 2 is one such method. Using this concept, we modify $\rho$ in terms of the count sketch matrix as shown below,

$$\rho \approx \rho R^{(k)^T} R^{(k)} = \tilde{\rho}R^{(k)}.$$

Each column of the count-sketch matrix $R^{(k)} \in \mathbb{R}^{c_k \times n}$ has a value of $\pm 1$ at a random row. Storing the count-sketch matrix in memory is not an overhead because of its inherent sparse nature. To illustrate, for the ogbn-products dataset, a single count-sketch matrix consumes approximately 88 MB of memory for a count-sketch ratio of 0.1. The count-sketch ratio's dependence on the approximation's quality is addressed in Lemma 4. However, an additional storage cost of $O(c_0 c_k)$ is incurred to store the sketches $\tilde{\rho}$ and $R^{(k)}$. In practise, the count-sketch matrices are sparse, with one non-zero entry per column. The update rule equation 4 is now modified as,

$$Z^{(l+1,k)} = \tilde{\rho}R^{(k)}\sigma\left(R^{(k)^T}\tilde{\rho}^T\left(\sum_q \left\{W^{(l,q)} - U\right\}\right) + T\right) - U \tag{5}$$

> **Challenge (2) Unsketching** of a matrix $G \in \mathbb{R}^{c_0 \times d_l}$ at layer $l$ from the sketch dimension $c_0$ to $n$ involves $O(c_0 n)$ computations and $O(n)$ memory.
>
> $$\textbf{unsketch(G)} = R^{(k)^T}\left(\tilde{\rho}^T G\right) + [\bar{x}]_{n \times d_l}$$

**Addressing challenge (2)**: The challenge at hand involves determining an efficient method for identifying active neurons at a layer $l$ without incurring the linear cost of computing all activations for a given input. This issue has been explored in previous literature (Chen et al. (2020), Chen et al. (2021)). The work presented in Chen et al. (2020) introduces an algorithm SLIDE which samples neurons at each layer, ensuring a sparse feed-forward propagation. SLIDE addresses this challenge by leveraging recent advancements in Maximum inner product search (MIPS) using asymmetric Locality-sensitive hashing (LSH) Shrivastava & Li (2014). The SLIDE algorithm Chen et al. (2020) introduces three parameters $(K, L, B)$, where $L$ denotes the number of hash tables and $K$ determines the number of hashcodes used for each hash table to select the bucket. $B$ denotes the input batch size. By utilizing a Locality-Sensitive Hashing algorithm parameterized by $K$ and $L$, along with MIPS hashing proposed by Shrivastava and Li in 2014 Shrivastava & Li (2014), a candidate set $S$ can be generated, where $|S| \leq \beta$. This approach incurs a one-time linear cost to preprocess the input set of vectors into hash tables. Subsequent adaptive sampling for a query $Q$ requires only a minimal number of hash lookups without the additional overhead of updating the hashtable weights, as the count sketches are fixed. The hash tables $HT$ are generated for the count sketch matrices $R^{(k)}$, and given a query which represents the node representations in the sketch space at layer $l$, activations are computed only for these active neurons in layer $l + 1$ given by set $S$. While LSH offers sublinear guarantees for approximate near-neighbour search, its query time

Table 1: Comparison of node representations between the PGCN method and the Taylor series approximation of the GCN update rule.

| Dataset | Method | $e^{(1)}$ Layer 1 | $e^{(2)}$ Layer 2 |
|---------|--------|---------|---------|
| Cora | PGCN | 0.7612 | 0.9096 |
| Cora | Taylor | 0.8487 | 1.1447 |
| Citeseer | PGCN | 0.81 | 1.1760 |
| Citeseer | Taylor | 0.9959 | 1.3237 |

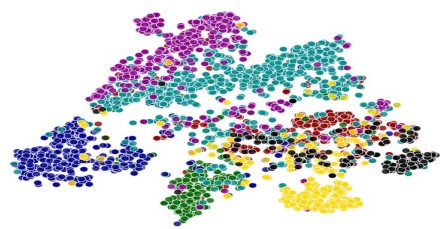

Figure 1: A t-SNE plot of the computed feature representations of the pre-trained PGCN at the first layer on the Cora dataset. Node colours denote classes.

efficiency is theoretically known to be inefficient due to the use of random hash functions. The core concept of employing LSH for adaptive sampling of neurons is elucidated in Chen et al. (2020). The paper discusses three strategies for sampling active neurons: 1) Vanilla sampling, 2) Top-$k$ sampling, and 3) Hard thresholding.

**Theorem 1** *Let $\mathcal{P}$ be the set of all orthogonal projection matrices of rank $c_0 < n$. The optimal projection matrix $Q \in \mathcal{P}$ for the update rule*

$$\tilde{X}^{(l+1)} = QC^{(l)}(\rho^T Z^{(0)} + [\bar{x}]_{n \times d_l})\Theta$$

*is identified as the POD projection matrix, which is expressed as $Q = \rho^T \rho$ (See proof in Appendix A).*

**Error bound on the node representations at each layer $l$.**

**Lemma 2** *Let $X^{(l)}$ and $\hat{X}^{(l)}$ represent the actual and approximate node representations for the PGNN method with the linearized GNN architecture at a layer $l$. Following the update rule $X^{(l+1)} = CX^{(l)}\Theta^{(l)}$, the normalized error $\epsilon^{(l+1)} = \frac{\left\| X^{(l+1)} - \hat{X}^{(l+1)} \right\|}{\left\| X^{(l+1)} \right\|}$ at layer $l+1$ caused by the PGNN method is given by,*

$$\epsilon^{(l+1)} \leq \left\| C - C_{eq} \right\| \left\| \Theta^{(l)} \right\| + \epsilon^{(l)} \left\| C_{eq} \right\| \left\| \Theta^{(l)} \right\| + \bar{T}$$

*where $\bar{T} = \left\| (I - P)[\bar{x}]_{n \times d_{(l+1)}} \right\|$ and the equivalent convolution matrix for the PGNN method $C_{eq} = PC, P = \rho^T \rho$. (See proof in Appendix A).*

Theorem 1 suggests that the POD method offers an optimal projection matrix for the Linearized GNN update rule. Lemma 2 indicates the quality of approximations made by the PGNN method depends on the equivalence of matrices $C$ and $C_{eq}$. The qualitative assessment of the learned feature representations can be conducted by visualizing the t-SNE transformed features from the first layer of a pre-trained PGCN model on the Cora dataset (Figure 1). The visualization reveals distinct clusters in the 2D projected space. These clusters align with the seven labels of the dataset, demonstrating the model's ability to distinguish between the seven topic classes in Cora effectively. Appendix D.1 is dedicated to empirical validation, wherein a series of experiments are conducted to ascertain the congruence of the convolution matrices for the Cora dataset.

## 4 RELATED WORK

The scalability of GNNs has been predominantly addressed through mini-batching strategies, which, despite mitigating memory bottlenecks, often fail to reduce epoch training time. Recent work in graph compression, such as Graph Coarsening Loukas (2018) and dataset condensation Zhao et al. (2021), aims for sublinear training times by condensing the graph, thus reducing node and edge counts Huang et al. (2021), Jin et al. (2022). These methods, however, face significant challenges:

the preprocessing overheads often exceed $O(n)$, reducing practical benefits, and the efficacy of the trained model varies with the GNN architecture used Jin et al. (2022), Ding et al. (2022). Scalable GNN approaches fall into several categories: (A) full-graph training, which is memory and time-intensive; (B) sampling-based methods like GraphSAGE Hamilton et al. (2018), FastGCN Chen et al. (2018b), and GraphSAINT Zeng et al. (2020), which employ various sampling strategies to reduce computational load; (C) historical-embedding methods, such as GNNAutoScale Fey et al. (2021) and VQ-GNN Ding et al. (2021), which store embeddings but incur high memory costs; (D) linearized GNNs Bojchevski et al. (2020), Wu et al. (2019), Frasca et al. (2020), which offer computational efficiency at the risk of oversimplification; (E) methods using random spanning trees Bonchi et al. (2024), which reduce computational load by transforming graphs into sparse path graphs; and (F) sketch-based methods like Sketch-GNN Ding et al. (2022), which approximate non-linear activations but struggle with error accumulation and high computational demands. Each approach presents trade-offs in terms of computational complexity and model expressiveness, addressing different constraints in GNN applications.

## 5 EXPERIMENTS.

In this section, we evaluate the proposed PGNN algorithm against existing graph compression techniques, including the Graph Coarsening approach (Coarsening Cai et al. (2021)) and the dataset condensation approach (GCond Jin et al. (2022)), both of which benefit from sublinear training time. Additionally, we compare PGNN with other sampling-based methods such as GraphSAINT Zeng et al. (2020), VQ-GNN Ding et al. (2021), and Sketch-GNN Ding et al. (2022). The performance of the PGNN framework is compared to these methods, with detailed experimental results provided in section 5. The graph datasets used for evaluation include $Cora$, $Citeseer$, $Pubmed$, $ogbn - arxiv$, $Reddit$, and $ogbn - products$. Furthermore, we present several ablation studies to analyse our method's effectiveness further. The PGNN update rules for various GNN architectures discussed in this section are detailed in Appendix B.

(1) **Evaluating the Quality of Node Representations at each Layer.** In Table 1, we compare the representations obtained without sketching (using the Taylor series) and with sketching, using the PGNN method, with the GCN architecture and it can be seen that error at the individual layers for the PGCN method is less when compared to the Taylor series method. The Taylor series approximation of the node representations and the node representations obtained from the PGCN method is utilized to compute $e^{(l)} \left( \frac{\|X^{(l)} - \hat{X}^{(l)}\|_F}{\|X^{(l)}\|_F} \right)$. For the first layer, we have:

$$X^{(1)} = \sigma(CX^{(0)}\Theta^{(0)}).$$

The Taylor series approximations of the node representations at layer one and $i-$th column are given by $\sigma(CX^{(0)}(\Theta^{(0)} + \Delta\Theta))_{:,i} = CX^{(0)}\Theta^{(0)}(:,i) + CX^{(0)}\Delta\Theta(:,i)$. The approximate node representations in the graph dimension $n$ by the PGCN method is $\hat{X}^{(l)} = \left\{ P\sigma \left( PC\hat{X}^{(0)}\Theta^{(0)} + [(I - P)\bar{x}]_{n \times d_1} \right) + [(I - P)\bar{x}]_{n \times d_1} \right\}.$

(2) **Comparing performance improvements obtained when using jumping knowledge networks.** As the depth of GNNs increases, there is a tendency for the node representations to converge to a standard value, a phenomenon called "over-smoothing" Li et al. (2018). A widely adopted mitigation approach in the literature is to bypass intermediate layers and directly contribute to the future layers by combining the Jumping Knowledge framework with models like GCN and GraphSAGE. Jumping Knowledge framework (Xu et al. (2018), Sun et al. (2024)) discusses various neighbourhood aggregation techniques and architecture changes, which help in finding better node representations and discuss theoretical guarantees about the general performance. In PGNN, as the depth of the GNNs increases, there is an accumulation of error, as shown for the linearized GNN in Lemma 2 affecting the downstream task. We use the skip-connections in the Jumping Knowledge framework as shown in Figure 5 while presenting the classification loss for the convergence aspect in Figure 4. Empirically, we find that the loss in accuracy due to depth for the Cora dataset was compensated by introducing skip connections as described in the Jumping Knowledge architecture in Figure 5. In the final layer of our model, we employed a layer aggregation technique. The layer aggregation process utilizes the formula

$$h_v^{final} = (Z^{(0)}(v, :), h_v^{(1)}, h_v^{(2)}, h_v^{(3)})\Theta_{\text{cat}}, \Theta_{\text{cat}} \in \mathbb{R}^{d_{eff} \times n_{\text{classes}}}$$

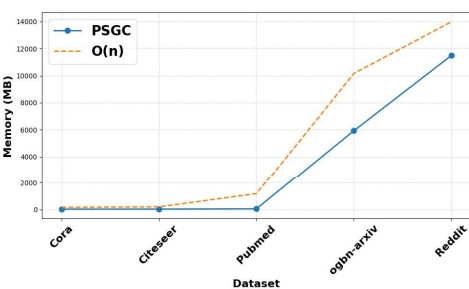

Figure 2: Memory complexity of the PSGC method.

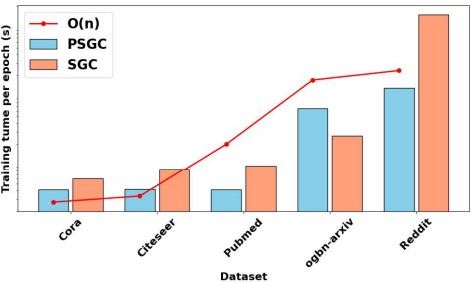

Figure 3: Training time comparison of the SGC and PSGC method.

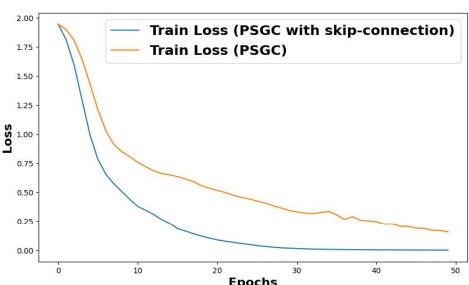

Figure 4: Classification loss when using Jumping knowledge network architecture on the PSGC versus the PSGC on a 3-layer GNN on the Cora dataset.

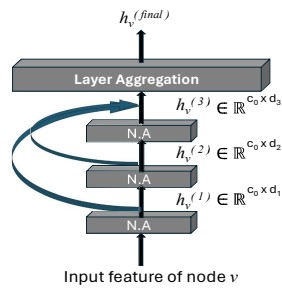

Figure 5: Jumping Knowledge network architecture for the PGNN method. N.A. denotes neighbourhood aggregation.

to effectively combine the information from the various layers. $n_{classes}$ denotes the number of output categories specific to the dataset. $d_{eff} = d_1 + d_2 + d_3$.

## 5.1 PROOF OF CONCEPT EXPERIMENTS

This section presents the experimental results of the PGNN framework, showcasing its performance on multiple datasets and its comparison with existing graph compression algorithms and sketch-GNN. The two-layer GNN setup of PGNN, with 128 hidden channels each, is evaluated using the update rule in equation 4 and the low-rank matrix decomposition Ye et al. (2016) of the matrix $\rho$. The evaluation, conducted on benchmark graph datasets, emphasizes classification accuracy. The efficiency of PGNN is compared with algorithms like Gcond Jin et al. (2022) and Graph Coarsening Cai et al. (2021). Detailed comparisons of node classification accuracies and sketch ratios across datasets provide insights into PGNN's efficiency in memory utilization and computational resources. Figures 2 and 3 highlight the sublinear memory complexity and training time complexities of PGNN, respectively, demonstrating its scalability and practical applicability. The results underscore the effectiveness and computational efficiency of the PGNN framework in graph-based learning tasks.

## 5.2 PERFORMANCE OF PGNN

We compare the PGNN framework's performance with Sketch-GNN and other sublinear training methods like graph coarsening Cai et al. (2021) and graph condensation Jin et al. (2022). The results, presented in Tables 2, 3 and 4, show that PGNN outperforms these methods, closely matching full-graph training performance on the Cora and Citeseer datasets with lower sketch ratios. GCond, while effective, has scalability issues and longer processing times, as seen with the Reddit dataset. When comparing PGNN across GNN architectures (SGC, GAT, and GraphSAGE) for the Cora and Citeseer datasets, we observe an accuracy loss, which is the same for the Pubmed, ogbn-arxiv and Reddit dataset (Tables 5, 6 and 4). In Table 6, we observe that PGNN, when applied to the ogbn-arxiv dataset, gave a 5% accuracy drop when using GraphSAGE on the full-graph and for the Reddit dataset a 0.9% accuracy drop with the sketch-GNN framework. There is however only an

accuracy drop of approximately 1% when using SGC on ogbn-arxiv (Table 2). We can observe that the training time per epoch for the Reddit dataset when using PSGC is less when compared to SGC method (Figure 3), which has a relative training time less than GCN, GraphSAGE and GAT Wu et al. (2019). With a low sketch ratio, PGNN maintains high classification accuracy, even outperforming SGC "Full-graph" on the Pubmed dataset (Table 2). As the graph size increases, the required sketch ratio to maintain full graph classification accuracy decreases. An important implementation aspect is the use of tensor sketch Pham & Pagh (2013) in the sketch-GNN framework, which has limitations with sparse tensors and less expressive power for non-linear activations. We address the propagation of error in deep GNNs and show that using the Jumping Knowledge framework Xu et al. (2018) on PGNN compensates for accuracy loss and ensures faster convergence.

## 5.3 EFFICIENCY OF PGNN

The comparative analysis reveals PGNN's efficiency through its notably reduced sketch ratio compared to established sketch-based methods like sketch-GNN, as evidenced in Table 2. For example, when examining the Reddit dataset with GCN architecture, while sketch-GNN required a sketch ratio of 0.3 to achieve a classification accuracy of around 92, PGNN achieved similar accuracy with a mere 0.05 sketch ratio. Importantly, PGNN eliminates the necessity for updating LSH hash tables, a process inherent in sketch-GNN. When evaluated against the ogbn-products dataset using the SGC architecture (refer to Table 2), PGNN experienced only a 1 percentage accuracy reduction. Moreover, despite PGNN's preprocessing time not being linear, it remains approximately one-sixth of the time consumed by graph compression algorithms like Gcond Jin et al. (2022).

Table 3: Performance comparison of the PGCN method on the Cora and Citeseer datasets with Sketch-GNN Ding et al. (2022), GCond Jin et al. (2022), Coarsening Cai et al. (2021). The sketch ratio for the PGCN method for Cora and Citeseer is kept at 0.02 and 0.018, while the results for the remaining methods are for a sketch ratio of 0.026.

| Benchmark | Cora | Citeseer |
|---|---|---|
| **Sketch-ratio** $(r = c_0/n)$ | 0.026 | 0.018 |
| **GNN Model** | **GCN** | |
| "Full-Graph" (Oracle) | $0.8119 \pm 0.0023$ | $0.7191 \pm 0.0018$ |
| Coarsening | $0.6518 \pm 0.0051$ | $0.5908 \pm 0.0045$ |
| GCond | $0.8002 \pm 0.0075$ | $0.7059 \pm 0.0087$ |
| Sketch-GNN | $0.8035 \pm 0.0071$ | $0.7114 \pm 0.0059$ |
| **PGCN** | $\mathbf{0.8039} \pm 0.0038$ | $\mathbf{0.7197} \pm 0.0004$ |

Table 2: Performance comparison of the PSGC and the SGC method with 2 layers. The classification accuracies of the SGC method are referenced from Wu et al. (2019) and Ding et al. (2022).

| Dataset | Nodes (n) | Edges (m) | Sketch ratio (r) | PSGC | SGC |
|---|---|---|---|---|---|
| **Cora** | 2,708 | 10,556 | 0.02 | $0.8019 \pm 0.0034$ | $0.81 \pm 0.00$ |
| **Citeseer** | 3,312 | 4,732 | 0.018 | $0.7174 \pm 0.002$ | $0.719 \pm 0.001$ |
| **Pubmed** | 19,717 | 44,338 | 0.01 | $0.7944 \pm 0.0013$ | $0.789 \pm 0.00$ |
| **ogbn-arxiv** | 169,343 | 1,166,243 | 0.06 | $0.6813 \pm 0.0017$ | $0.6944 \pm 0.0005$ |
| **Reddit** | 232,965 | 114,615,892 | 0.05 | $0.9272 \pm 0.0006$ | $0.9464 \pm 0.0011$ |
| **ogbn-products** | 2,449,029 | 61,859,140 | 0.001 | $0.6564 \pm 0.00$ | $0.6638 \pm 0.0029$ |

Table 5: Performance comparison of the PSGC, PGCN, PSAGE with a sketch ratio $r = 0.01$ with GCN, GAT, and other standard GNN architectures for the Pubmed dataset.

| Method | GCN | GAT | FastGCN | GIN | SGC | PSGC | PGCN | PSAGE |
|---|---|---|---|---|---|---|---|---|
| **Pubmed** | $79.0 \pm 0.4$ | $78.5 \pm 0.3$ | $77.4 \pm 0.3$ | $77.0 \pm 1.2$ | $78.9 \pm 0.0$ | $\mathbf{79.44 \pm 0.1}$ | $77.3 \pm 0.1$ | $76.1 \pm 0.1$ |

Table 4: Comparison of PSGC, PGAT, PSAGE method with Gcond Jin et al. (2022) for Cora and Citeseer.

| Dataset | Methods | SGC | GAT | SAGE |
|---|---|---|---|---|
| **Cora** | Gcond ($r = 0.026$) | 76.1 | - | 76.0 |
| | **PGNN** ($r = 0.02$) | **80.19 ± 0.34** | 77.1 ± 0.0 | 78.42 ± 0.33 |
| **Citeseer** | Gcond | 71.6 | - | 69.2 |
| | **PGNN** ($r = 0.018$) | **71.7 ± 0.7** | 70.2 | 69.2 |

Table 6: Performance comparison of PGNN method with Graph-SAINT Zeng et al. (2020), VQ-GNN Ding et al. (2021), sketch-GNN Ding et al. (2022), Graph Coarsening Cai et al. (2021) and linearized GNN (SGC Wu et al. (2019)) on Reddit and ogbn-arxiv.

| Benchmark | ogbn-arxiv | | Reddit | |
|---|---|---|---|---|
| **SGC** | $0.6944 \pm 0.0005$ | | $0.9464 \pm 0.0011$ | |
| **GNN Model** | GCN | GraphSAGE | GCN | GraphSAGE |
| **"Full-Graph" (Oracle)** | $0.7174 \pm 0.0029$ | $0.7149 \pm 0.0027$ | - | - |
| **Graph-SAINT** | $0.7079 \pm 0.0057$ | $0.6987 \pm 0.0039$ | $0.9225 \pm 0.0057$ | $0.9581 \pm 0.0057$ |
| **Coarsening** | $0.6892 \pm 0.0035$ | $0.7048 \pm 0.0080$ | - | - |
| **VQ-GNN** | $0.7055 \pm 0.0033$ | $0.7028 \pm 0.0047$ | $0.9399 \pm 0.0021$ | $0.9449 \pm 0.0024$ |
| **Sketch Ratio** ($r = c/n$) | $r = 0.4$ | | $r = 0.3$ | |
| **Sketch-GNN** | $0.7028 \pm 0.0087$ | $0.7048 \pm 0.0080$ | $0.9280 \pm 0.0034$ | $0.9485 \pm 0.0061$ |
| **PGNN** | $r = 0.06$ | | $r = 0.05$ | |
| | $0.6785 \pm 0.0020$ | $0.66 \pm 0.0050$ | $0.9271 \pm 0.0006$ | $0.9302 \pm 0.0014$ |

## 6 CONCLUSION

In conclusion, we present PGNN, a novel sketch-based framework for addressing graph-based learning tasks. Through proof-of-concept experiments and comparative analyses, we have demonstrated PGNN's capability to achieve competitive classification accuracies with significantly reduced sketch ratios compared to established sketch-based methods. Furthermore, the sublinear memory complexity and running time of PGNN underscore its scalability and practical applicability in efficiently managing large-scale graph data. The results presented in this work offer valuable insights into the potential of PGNN to streamline graph-based learning processes, mitigating challenges posed by memory limitations and computational resources. Future research directions include adapting the PGNN framework to streaming graph scenarios and developing efficient techniques for creating synthetic bases for sketch matrices in graph learning.

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
