# SUPPLEMENTARY MATERIAL FOR PROPER ORTHOGONAL DECOMPOSITION FOR SCALABLE TRAINING OF GRAPH NEURAL NETWORKS

## A    APPENDIX

**Definition 1** *The column space of two matrices $A$ and $B$ are the same if there exists a matrix $M$ such that $A = BM$.*

### A.1    PROOF FOR LEMMA 2

Let the equivalent node representations at layer $l+1$ in the graph dimension $n$, be $\hat{X}^{(l+1)}$, the PSGC update rule (Appendix B.1) is given by the following expression:

$$\hat{X}^{(l+1)} = PC\hat{X}^{(l)}\Theta^{(l)} + (I - P)[\bar{x}]_{n \times d_{(l+1)}},$$

$$\epsilon^{(l+1)} = ||CX^{(l)}\Theta^{(l)} - (PC\hat{X}^{(l)}\Theta^{(l)} + (I - P)[\bar{x}]_{n \times d_{(l+1)}})||,$$

$$\epsilon^{(l+1)} = ||CX^{(l)}\Theta^{(l)} - C_{eq}X^{(l)}\Theta^{(l)} + C_{eq}X^{(l)}\Theta^{(l)}$$

$$- (C_{eq}\hat{X}^{(l)}\Theta^{(l)} + (I - P)[\bar{x}]_{n \times d_{(l+1)}})||,$$

Using the triangular inequality and sub-multiplicative property of norms,

$$\epsilon^{(l+1)} \leq ||C - C_{eq}||\ ||\Theta^{(l)}|| + \epsilon^{(l)}\ ||C_{eq}||\ ||\Theta^{(l)}|| + \bar{T}.$$

**Lemma 3** *Columnspace of matrix $B = C^{(l)}\rho^T$ is equal to the columnspace of matrix $\rho^T$.*

*Proof:* From Definition 1, If the column-space of $B$ and column-space of $\rho^T$ are similar, then there exists a matrix $M$ such that $\rho^T M = B$. The matrix $M$ which accomplishes this $M = \rho B$ ($\rho\rho^T = I_{c_0}$). This shows that matrix $B$ and $\rho^T$ have the same column space.

Lemma 3 implies that for an arbitrary vector $z \in \mathbb{R}^{c_0}, C\rho^T z$, it can be represented as $\rho^T \alpha$, where $\alpha \in \mathbb{R}^{c_0}$. Also since the matrix $\rho^T$ denote the left singular vectors of the SVD of the matrix $\frac{\tilde{X}^{(0)} - [\bar{x}]_{n \times c_0}}{\sqrt{nc_0 - 1}}$, they span the column space of the matrix $F = \frac{\tilde{X}^{(0)} - [\bar{x}]_{n \times c_0}}{\sqrt{nc_0 - 1}}$. Thus $C\rho^T z = \sum_{i=1}^{c_0} \omega_i F(:,i)$ for arbitrary values of $\omega_i$. The matrix $\tilde{X}^{(0)}$ denotes the augmented input node feature matrix.

### A.2    PROOF FOR THEOREM 1

The output layer node representations for the linearized GNN using the optimal approximation of the input node feature matrix given by the POD,

$$\hat{X}^{(l)} = C^{(l)}\rho^T Z^{(0)}\Theta + [C^{(l)}\bar{x}]_{n \times d}\Theta = B\Psi + \Psi_1,$$

$B = C^{(l)}\rho^T, \Psi = Z^{(0)}\Theta, \Psi_1 = [C^{(l)}\bar{x}]\Theta$. Consider $d$ as the number of hidden channels at layer $l$. We propose that the projection matrix, denoted by $Q$, is the product of two matrices $Q_1$ and $Q_2$, such that $Q = Q_1 Q_2$. In this context, $Q_2$ signifies the linear transformation responsible for reducing the dimensionality of matrices. Conversely, $Q_1$ is the matrix that handles the inverse projection. The objective is now to find

$$\arg\min_Q ||QB\Psi - B\Psi||$$

$$\arg\min_Q \sum_{i=1}^{c} ||QB\Psi(:,i) - B\Psi(:,i)||$$

$$\arg\min_Q \sum_{i=1}^{c} \left\|\sum_{j=1}^{c} Q\omega_{ji}F(:,j) - \omega_{ji}F(:,j)\right\| \quad \text{(Lemma 3)} \tag{6}$$

$$\arg\min_Q \sum_{i=1}^{c} \left\|\sum_{j=1}^{c} \frac{\bar{\omega}_{ji}}{p_2}(Q\tilde{X}^{(0)}(:,j) \quad -\tilde{X}^{(0)}(:,j)) - \bar{\omega}_{ji}\frac{Q(p_3) - p_3}{p_2}\right\|$$

$p_3 = \sum_{i=1}^{c_0} \frac{\tilde{X}^{(0)}(:,i)}{nc_0}$. We denote $\mathcal{P}$ as the manifold of all rank $(c_0 < n)$ orthogonal projection matrices of size $n \times n$. The manifold is referred to as Grassmannian in the literature. The POD projection matrix $P$ is a minimizer of the function $e$,

$$e(P, x) = <Px - x, Px - x> \quad \text{where } x \in \tilde{X}^{(0)} \text{ (See Rathinam \& Petzold (2003))}.$$

This, in turn, proves that the best matrix $Q$ which can be used in the scenario is the POD projection matrix $P$.

**Lemma 4 (Ding et al. (2022))** *Given matrices $C \in \mathbb{R}^{n \times n}$ and $(X^{(l)}W^{(l)})^T \in \mathbb{R}^{d \times n}$, consider a randomly selected count sketch matrix $R \in \mathbb{R}^{c_k \times n}$ (defined in section 2), where $c_k$ is the sketch dimension, and it is formed using $r = \sqrt{jn}$ underlying hash functions drawn from a 3-wise independent hash family $\mathcal{H}$ for some $j \geq 1$. If $c_k \geq \frac{2+3j}{\varepsilon^2 \delta}$, we have*

$$\Pr\{\|(CR_k^T)(R_k X^{(l)}W^{(l)}) - CX^{(l)}W^{(l)}\|_F^2 > \varepsilon^2 k\|C\|_F^2\|X^{(l)}W^{(l)}\|_F^2\} \leq \delta.$$

# B    GENERALIZE TO MORE GNNs

This section presents a compendium of prevalent GNNs that can be tailored to fit into the unified framework delineated in section 2. The crux of most GNN architectures revolves around message passing among node features, followed by feature transformation and activation functions—a process commonly known as 'generalized graph convolution'. Within this overarching framework, the distinctions among GNNs primarily arise from their choice of convolution matrices, denoted as $C^{(q)}$, which can either remain static or evolve as trainable parameters. A trainable convolution matrix is contingent upon input data and adjustable parameters, potentially varying across different layers, as denoted by $C(l,q)$.

$$C_{i,j}^{(l,q)} = \underbrace{\mathcal{C}_{i,j}^{(l,q)}}_{\text{fixed}} \cdot \underbrace{h_{\theta^{(l,q)}}^{(q)}(X_{i,:}^{(l)}, X_{j,:}^{(l)})}_{\text{learnable}}$$

We analyze how the PGNN framework works with various GNN architectures. The architecture of focus involves SGC Wu et al. (2019), GCN Kipf & Welling (2017), SAGE Hamilton et al. (2018), GAT Veličković et al. (2018). In the below subsections, we discuss how the PGNN framework works with various GNN architectures using update rules as described in the supplementary material of Ding et al. (2022). We present two theorems from Ding et al. (2021), which can be crafted to the PGNN framework. The theorems illustrate how node representations and gradients during backpropagation can deviate when using the PGNN method compared to the conventional update rules of the GNN. The upper bound for the error at a specific layer $l$ $(\epsilon^{(l)})$ is provided in Lemma 2.

**Theorem 5 (Ding et al. (2021))** *If the relative error of the l-th layer for the PGNN method is $\varepsilon^{(l)}$, the convolution matrix $C^{(l)}$ is either fixed or learnable with the Lipschitz constant of $h_\theta^{(l)}(\cdot) : \mathbb{R}^{2f_l} \to \mathbb{R}$ upper-bounded by $Lip(h_\theta^{(l)})$, and the Lipschitz constant of the non-linearity is $Lip(\sigma)$, then the estimation error of forward-passed features satisfies,*

$$\|\hat{X}^{(l+1)} - X^{(l+1)}\|_F \leq \varepsilon^{(l)} \cdot (1 + O(Lip(h_\theta^{(l)})))Lip(\sigma)\|C^{(l)}\|_F\|X^{(l)}\|_F\|W^{(l)}\|_F.$$

**Theorem 6 (Ding et al. (2021))** *If the conditions in Theorem 5 hold and the non-linearity satisfies $|\sigma'(z)| \leq \sigma'_{max}$ for any $z \in \mathbb{R}$, then the estimation error of back-propagated gradients satisfies,*

$$\|\hat{\nabla}_{X^{(l)}}\ell - \nabla_{X^{(l)}}\ell\|_F \leq \varepsilon^{(l)} \cdot (1 + O(Lip(h_\theta^{(l)}))\sigma'_{max}\|C^{(l)}\|_F\|\nabla X^{(l+1)}\|_F\|W^{(l)}\|_F.$$

## B.1    PGNN WITH SGC.

The node representation at layer $l+1$, $Z^{(l+1)}$ given by the PGNN method is $Z^{(l+1)} = S_C Z^{(l)}\Theta^{(l)} + [\rho C \bar{x}]_{n \times d_l}\Theta^{(l)}$. The sketch of the convolution matrix $S_C = \rho C \rho^T$.

## B.2 PGNN WITH GCN.

Let $W^{(l)} = S_C Z^{(l)} + [\rho C \bar{x}]_{n \times d_l}$. The update rule of PGCN is given by $Z^{(l+1,k)} = \tilde{\rho}^{(k)} R^{(k)} \sigma(\textbf{unsketch}\left\{(W^{(l)})\Theta^{(l)}\right\}) - [\rho \bar{x}]_{c \times d_{l+1}}$. Detailed steps for the PGCN method are explained in Algorithm 1.

## B.3 PGNN WITH SAGE.

With $U = [\rho \bar{x}]_{n \times d_{l+1}}, S_{C^{(1)}} = I_{c \times c}$ and $S_{C^{(2)}} = \rho C^{(2)} \rho^T$. The update rule for PSAGE is given by

$$Z^{(l+1,k)} = \tilde{\rho}^{(k)} R^{(k)} \sigma(\textbf{unsketch}\{S_{C^{(1)}} Z^{(l)} W^{(l,1)} + [\rho \bar{x}]_{n \times d_l} W^{(l,1)} + S_{C^{(2)}} Z^{(l)} W^{(l,2)} +$$

$$[\rho C^{(2)} \bar{x}]_{n \times d_l} W^{(l,2)} - U\}) - U.$$

The identity matrix, the first convolution for the SAGE architecture, is converted to the identity matrix in the sketch dimension because of the orthonormal property of the eigenvectors. The sketched second convolution matrix is given by $\rho C^{(2)} \rho^T$.

## B.4 PGNN WITH GAT.

The convolution mechanism intrinsic to the GAT architecture is inherently learnable. While PGAT offers a memory benefit during the forward pass through LSH, it necessitates the transfer of the $n \times n$ convolution matrix to the device, thereby limiting the overall memory efficiency.
$\tilde{E}^{(l,q)} = V^{(l,q)} + V^{(l,q)^T}, \quad V^{(l,q)} = (\rho^T Z^{(l,q)} + [\bar{x}]_{n \times d_l})\Theta^{(l,q)} a^{(l,q)},$
$C = \mathcal{A} + I, \quad a^{(l,q)} \in \mathbb{R}^{d_{l+1}}, C^{GAT} = C \odot \exp(\text{LeakyReLU}(\tilde{E}^{(l,q)})).$

$$Z^{(l+1,q,k)} = \tilde{\rho}^{(k)} R^{(k)} \sigma\left(\textbf{softmax}\left(C^{GAT}\right) \textbf{unsketch}\left(Z^{(l,q)}\right) \Theta^{(l,q)}\right). \tag{7}$$

A subset $\beta$ number of nodes are unsketched at each layer using the LSH method based on MIPS.

## B.5 ALGORITHM

Algorithm 1 details the procedural framework of the PGNN, which is executed within the structure of the GCN architecture. The algorithm can be generalized to the architectures discussed in sections B.1 and B.3. However, for the PGAT architecture (See section B.4), the computation of the sketch for the convolution matrix must be omitted.

# C COMPLEXITY ANALYSIS

We delineate the intricacies inherent in the algorithm with the PGNN framework.

**One-time Preprocessing:** The pre-processing step involves finding the right singular vectors of the matrix described in Algorithm 1, which takes time $O(dnc_0)$. Computing $S_C = \rho C \rho^T$ takes time $O(n^2)$. Computing the sketch of the initial node feature matrix $S_X = \rho(X^{(0)} - [\bar{x}]_{n \times d})$ takes time $O(nc_0 d)$. Computing the sketches of the matrix $\rho$ to obtain $\tilde{\rho}$ has linear time complexity. The pre-processing phase has a time complexity of $O(n^2)$ and a memory complexity of $O(n)$.
**Overhead of computing LSH hash tables.** The time complexity for computing the hash index for each node is $O(c_0 c_k)$ when using Simhash (See section 2), and since there are $n$ nodes and $r$ hash tables, we get an overhead of $O(rnc_0 c_k)$ for time and $O(rc_0 c_k)$ for space.
**Training complexities with PGNN.** We present the complexities within the context of the GCN architecture. Forward and backward pass: involves $O(c_0 d^2 L) + O(c_k c_0 dL) + O(\beta c_k c_0 L)$, which reduces to $O(c_k c_0)$ time. The memory complexity in the backward propagation is $O(c_0) + O(c_k) + O(\beta)$. The third term $O(\beta)$ occurs when using the vanilla sampling for LSH. The memory complexity involved is $O(\beta dL + d^2 L)$.
**Inference**: incurs $O(n)$ time and $O(n)$ memory as is the case in a standard GCN.
**Remark.** The training complexities mentioned above do not hold for the GAT architecture Veličković et al. (2018) because of the inherent nature of the operations involved, which is expounded in Appendix B.4. The underlying complexities in the original GAT architecture will hold, and for completeness, we present the accuracies for the Cora and Citeseer datasets using PGAT in Table 4.

---

**Algorithm 1** PGNN

---

**Require:** Node feature matrix $X^{(0)} \in \mathbb{R}^{n \times d}$, labels $y$, Convolution matrix $C$, sketch ratio $r = \frac{c_0}{n}$, $k$-number of sketches, count sketch matrix dimension $c_k$, $\beta$.
 1: **Preprocessing step:**
 2: Compute sketch dimension $c_0 = \lceil rn \rceil$.
 3: **if** $c_0 > d$ **then**
 4:     Generate random vectors $r_1$ and $r_2$ of size $c_0 - d$, where each element $r_1(j)$ and $r_2(j)$ is drawn from the set $\{1, 2, \dots, d\}$.
 5:     Produce augmented feature matrix $\tilde{X}^{(0)}$. $\tilde{X}^{(0)} = [X^{(0)} M]$. $M = X^{(0)}(:, r_1) \odot X^{(0)}(:, r_2)$.
 6: **end if**
 7: Compute mean vector $\bar{x} = \frac{\sum_{i=1}^{c_0} \tilde{X}^{(0)}(:, i)}{c_0}$.
 8: Obtain left singular vectors $\rho^T$ of the matrix $\frac{\tilde{X}^{(0)} - [\bar{x}]_{c_0 \times d}}{\sqrt{c_0 d - 1}}$ Halko et al. (2010).
 9: Sketch input node feature matrix $Z^{(0)} = \rho(X^{(0)} - [\bar{x}])$.
10: Project convolution matrix to obtain $S_C = \rho C \rho^T$, compute vectors $\rho \bar{x}$, $\rho C \bar{x}$.
11: Generate $k$ count-sketch matrices $R^{(1)}, \dots, R^{(k)} \in \mathbb{R}^{c_k \times n}$ and obtain sketches $\tilde{\rho}^{(1)}, \dots, \tilde{\rho}^{(k)} = \rho R^{(k)^T}$.
12: **Training of GNN:**
13: Initialize GNN weights randomly.
14: **for** epoch = 1,2, to ... **do**
15:     **for** $l$ = 1 to $L$ **do**
16:         Follow forward propagation rule to obtain sketched layer representation $Z^{(l)}$ using propagation rule (Appendix B.2). Unsketching of an arbitrary matrix $F$ at layer $l$, **unsketch**$(F) = \tilde{\rho}^{k^T}(S, :)F + [\bar{x}]_{n \times d_l}$ for a subset $S$ containing almost $\beta$ number of nodes, using LSH MIPS.
17:         Update GNN weights for each layer $l$ using the loss $\ell$.
18:         Backpropagate and update weights $\Theta^{(l)}$.
19:     **end for**
20:     **return** Learned weights $\Theta^{(l)}, l = 1,2, \dots, L$
21: **end for**
22: **Inference:**
23: Predict using standard GCN update rule with learned weights $\Theta^{(l)}, l = 1,2, \dots, L$.

---

**An implementation detail.** When the sketch ratio $r$ is such that $\lceil rn \rceil > d$, which is the feature dimension, the PCA or the POD method necessitates computing the covariance matrix Ding et al. (2021). To overcome the challenge of storing and computing the covariance matrix, we use the feature engineering method to augment $X^{(0)}$ by selecting random combinations of columns of this matrix to find the augmented input node feature matrix $\tilde{X}^{(0)}$ (See Algorithm 1).

# D   ADDITIONAL EXPERIMENTS

## D.1   COMPARISON OF SPECTRAL PROPERTIES OF THE SKETCHES OF THE CONVOLUTION MATRICES

We say that a matrix $B \in \mathbb{R}^{n \times n}$ is an $\epsilon$ approximation to matrix $A \in \mathbb{R}^{n \times n}$ if their quadratic forms have the form

$$\frac{x^T B x}{\epsilon} \leq x^T A x \leq \epsilon\, x^T B x \ \ \forall\, x \in \mathbb{R}^n.$$

The above equivalence implies the spectrum similarity between the two matrices (Courant-Fisher Theorem Chung (1997)). We present comparisons of the eigenvalues and eigenvectors of the convolution matrix $C$ and the equivalent convolution matrix $C_{eq} = PC$ for the Cora dataset in Figures 6a and 6b. The eigenvalues and eigenvectors of the matrix $C_{eq}$ closely align with those of $C$.

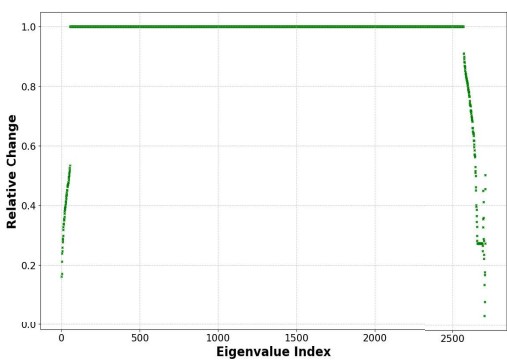
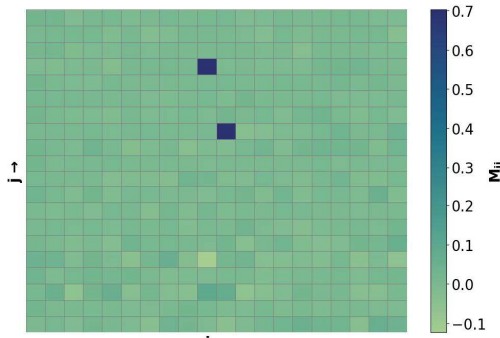

(a) Eigenvalue comparison of matrices $C$ and $C_{eq}$ for the Cora dataset. Here the relative change is given by $\frac{|\lambda_i(C) - \lambda_i(C_{eq})|}{\lambda_i(C)}$. $\lambda_i(C)$ denoting the $i-th$ eigenvalue of $C$.

(b) Comparison of Eigenvectors of $C$ and $C_{eq}$ for the Cora dataset, with the $ij$-th element $M_{ij}$ quantified by $\phi_{ij}(C) - \tilde{\phi}_{ij}(C_{eq})$, where $\phi_{ij}(C)$ and $\tilde{\phi}_{ij}$ denote the $ij$-th component of the eigenvectors for $C$ and $C_{eq}$, respectively.

Figure 6: Comparison of eigenvalues and eigenvectors of $C$ and $C_{eq}$ for the Cora dataset.

| Dataset | Cora | Citeseer | Pubmed | ogbn-arxiv | Reddit |
|---|---|---|---|---|---|
| Task | node | node | node | node | node |
| Setting | transductive | transductive | transductive | transductive | transductive |
| Label | single | single | single | single | single |
| Metric | accuracy | accuracy | accuracy | accuracy | accuracy |
| # of Nodes | 2708 | 3327 | 19717 | 169,343 | 232,965 |
| # of Edges | 5429 | 4732 | 44338 | 1,166,243 | 11,606,919 |
| # of Features | 1,433 | 3,703 | 500 | 128 | 602 |
| # of Classes | 7 | 6 | 3 | 40 | 41 |

Table 7: Detailed Overview of the graph datasets utilized in experiments.

# E  IMPLEMENTATION DETAILS

We outline the various implementation details with the hyper-parameter setups for experiments in section 5.

**Datasets.** Table 7 provides a comprehensive summary of the statistics for all datasets utilized in the experiments. The datasets ogbn-arxiv and ogbn-products were sourced from the Open Graph Benchmark (OGB)[1]. The Reddit dataset, a more streamlined variant of the original dataset by Hamilton and colleagues, was acquired through the PyTorch Geometric library[2]. For our research, we adhered to the conventional dataset divisions established by OGB and PyTorch Geometric.

**Code Frameworks.** The codes used for experimentation are made available at repository[3]. PGNN framework make use of the PyTorch library and the PyTorch Sparse library[4]. For the computation of the sketch of the input node feature matrix, the svd function from the Pytorch library is used. We provide proof of concept results for the Citeseer dataset with the optimum hyper-parameters in the repository 3 where the Count-sketch technique implementation is taken from the repository[5] and the LSH hashing and query function implementations are taken from repository[6]. All of the above code repositories we used are licensed under the MIT license.

**Computational Infrastructures.** All of the experiments are conducted on the Nvidia A30 GPU with Xeon CPUs.

---

[1] https://ogb.stanford.edu/

[2] https://github.com/pyg-team/pytorch_geometric

[3] https://anonymous.4open.science/r/POD-Scalable-training-for-GNNs-63F8

[4] https://github.com/rusty1s/pytorch_sparse

[5] https://github.com/johnding1996/Sketch-GNN-Sublinear

[6] https://github.com/keroro824/HashingDeepLearning

**GNN training:** For all the datasets, we use a 2-layer GNN with the number of hidden channels kept at 128; the learning rate used is changed for different datasets and different GNN architectures. Dropout and batch-norm are not used, and Adam is the default optimizer. The default learning rate is 0.001.

**Setup of PGNN:** In our experimental setup, we have designated 500 epochs for each run, with 10 runs to ensure statistical significance. The sketch-ratio of 0.018 used for the citeseer is same as mentioned from the paper Ding et al. (2022), sketch-ratio of 0.02 against 0.026 is used for the cora dataset to demonstrate the effectiveness of the proposed method. For the Pubmed dataset, we selected a lower sketch ratio of 0.01. This choice aligns with the general principle that as graph size increases, the sketch ratio or the effective number of components for preserving variance decreases. Extensive experimentation confirmed that a sketch ratio of 0.01 was sufficient to achieve good classification accuracy. For the ogbn-arxiv, Reddit, ogbn-products datasets, although the standard ratio is 0.4, 0.3, 0.2, our experiments indicated that a lower ratio of 0.06, 0.05, 0.001 provided a better balance between computational efficiency and model accuracy. We observe that the time taken for a single query when using the implementation of LSH from repository 6 for the Cora dataset is in the range of 0.0028-0.017 seconds which affects training time. For proof of concept experimental results described in section 5, we use the update rule in equation 4 employing the low-rank approximation method for $\rho$ using SVD for Pubmed, ogbn-arxiv and Reddit dataset, with rank parameter as min($\lceil rn \rceil$), 12000). For PGAT, we employ 2 attention masks. For experiments validating the update rule in equation 5 with Count-sketch, we set the ratio $c_k$ to $\lceil 0.5n \rceil$ and used 2 sketches. In the case of LSH, we configured the number of hash functions $K$ to 5 and the number of hash tables $L$ to 6. The training times of PGNN were not compared with the existing sketch-based method, Sketch-GNN Ding et al. (2022), due to observed discrepancies in node classification accuracy from the implementation available in the repository 5. Additionally, the implementation of Sketch-GNN for the ogbn-arxiv, Reddit and the ogbn-products dataset encountered memory issues during experimentation on the Nvidia A30 GPU.