# OpenReview forum: "Proper Orthogonal Decomposition for Scalable Training of Graph Neural Networks"
_ICLR.cc/2025/Conference — ICLR 2025 Conference Withdrawn Submission_

### Official Review · Reviewer_9WvS · 2024-11-04

**Soundness:** 3
**Presentation:** 2
**Contribution:** 1
**Rating:** 3
**Confidence:** 5

**Summary:**

The authors propose an algorithm to shrink the memory and computational complexity of forward passes in a message-passing based setting for GNNs. The authors rely on a proper orthogonal decomposition (POD) to achieve compression, mixed with some classical sketching techniques such as locality-sensitive hashing and count sketches. The method is able to achieve high compression over some graph datasets while remaining competitive against the uncompressed setting and baselines such as Sketch-GNN. Theory is provided on the optimality of the POD along with error bounds on the approximation.

**Strengths:**

-The method is able to achieve a high-degree of compression on ogbn-arxiv and reddit with comparable accuracy with compared against sketch-gnn which is using less-aggressive compression.

-The POD ostensibly has not been considered in this type of training regime before.

-Wide set of experiments -- citeseer, cora, ogbn-arxiv, reddit are all classical GNN datasets along with comparisons against sketch-gnn, Graph-SAINT, VQ-GNN.

**Weaknesses:**

-Novelty is limited: the approach is simply using a low-rank projector and still relies on count-sketches per the update rule (5). The sketched update rules look remarkably similar to those of Sketch-GNN minus the POD which is taking care of the non-linearity (more on this below).

-Sketch-GNN uses polynomial tensor sketches to avoid blow-up in the backwards pass when examining the derivative with respect to the activation function. The authors of this work are using LSH (per SLIDE) to avoid this. A few issues with this: (1) Is this really less complicated or computationally-less efficient than using learnable sketches? LSH (the SimHash) relies on dense Gaussian matrices and even SLIDE acknowledges learnable projections must be used. (2) The usage is not appropriate: observe that the LSH in SLIDE is used in the *final* layer as the magnitude of the dot product directly corresponds to the magnitude of the logit, i.e., class probability. When used in intermediate layers, all you are doing is ignoring smaller activations -- but these can be very important in the update, which is why the SLIDE strategy is nearly exclusively used for the final layer of massive, extreme multi-label compression tasks. (3) Sneaking in LSH for optimized forward and backwards passes is a non-trivial engineering task. The audience would appreciate seeing computational run-times associated with this overhead.

-The theory is weak. In Theorem 1, the authors should clarify what they mean by "optimal projection matrix" as you have to look in the Appendix to gather it. The result is a close cousin of the Eckhart-Young-Mirsky theorem and the result is a few obvious inequality simplifications followed by a citation. The appendix recycles several lemmas from Ding et al., 2022 and the error bound, again, follows routine calculations from sketching theory.

-The experimental results are weak. Table 3 shows the results are within-error equivalent to Sketch-GNN thus showing non-trivial improvement. In Table 4, the accuracies mostly lag or minimally improve accuracy. The authors should increase the sketch-ratio until the accuracy exceeds their competitors so the audience can understand the performance curves better. Table 6 has the same issue -- just increase the ratio until the PGNN outcompetes Sketch-GNN so we can understand at which ratio this will occur.

Minor: Please fix the citations. They read as normal text -- parenthesize them.

**Questions:**

See weaknesses.

---

### Official Review · Reviewer_fsuo · 2024-11-04

**Soundness:** 2
**Presentation:** 2
**Contribution:** 1
**Rating:** 3
**Confidence:** 5

**Summary:**

The paper proposes approximating the feature and the adjacency matrices into a lower dimensional subspace to increase computational efficiency of model training. Theorems about the quality of the approximations are proved. Experiments show how effective the proposed approach is in terms of trained model quality.

**Strengths:**

* The proposed method makes GNN training a little bit more efficient.
* The theorems make the proposed method theoretically sound.

**Weaknesses:**

* The speedup one gets by sacrificing model quality is not great.
* For a proposed method claiming to improve efficiency, the datasets used are small. ogbn-papers100M, ogbn-mag240M, igb-het, igb-hom datasets would be more appropriate to prove the real worth of the method.
* Figure 3 y-axis has no reference numbers.
* Experiments against GraphBolt's [1] fully GPU-enabled GraphSAGE implementation [2] should be made if the authors want to compare against one of the best available GraphSAGE implementations, when it comes to runtime efficiency.
* The experimental evaluation is focused against the SGC baseline, reducing the impact of the work. When the method is compared against nonlinear baselines, it does not fare well (Table 6).


[1] https://www.dgl.ai/dgl_docs/api/python/dgl.graphbolt.html

[2] https://github.com/dmlc/dgl/blob/master/examples/graphbolt/pyg/node_classification_advanced.py

**Questions:**

* I would expect the use of billion scale datasets instead of datasets such as Cora and Citeseer, which are not reliable datasets to compare model quality. Do the authors have any results for such datasets?

---

### Official Review · Reviewer_kYJy · 2024-11-05

**Soundness:** 2
**Presentation:** 2
**Contribution:** 2
**Rating:** 3
**Confidence:** 4

**Summary:**

This paper proposes a sketch-based GNN training method that does not require updating the sketches during training. PGNN uses POD sketches to approximate the update rules of GNNs. The effectiveness of the PGNN method is validated by experimental results on five datasets.

**Strengths:**

1. This paper is easy to understand.

2. The idea is novel.

**Weaknesses:**

1. As a work on GNN training, the current amount of experimentation is far from sufficient. The authors only presented very few experimental results, which greatly undermines the solidity of this paper. The results from Table 2 to Table 6 are scattered, missing many important results, such as the results of GCN and GAT on Products.
2. There seems to be a problem with the format of this paper. I am unable to select any text.
3. The author mentions in Section 3 that the proof can be found in the appendix A, but I was unable to find the appendix A.

**Questions:**

See weakness

---

### Official Review · Reviewer_Vf5d · 2024-11-09

**Soundness:** 3
**Presentation:** 1
**Contribution:** 2
**Rating:** 5
**Confidence:** 3

**Summary:**

This paper aims to develop a scalable training approach for GNNs based on the proper orthogonal decomposition (POD) technique. It proposes the PGNN approach, which includes a preprocessing stage to sketch the input node feature matrix, sketch the convolution matrix, and generate count-sketch matrices to obtain the sketches. Then by using the sketches in the training stage, it greatly reduces the complexity as the dimensionality is reduced. The paper provides theoretical justification as well as the empirical study results to justify the power of PGNN.

**Strengths:**

1. Enhancing GNN efficiency is an important topic.

2. Theoretical justification is provided to demonstrate how POD can preserve graph information.

3. Using POD to improve GNN efficiency is a novel approach.

**Weaknesses:**

1. Clarity - The paper is not very easy to follow.
- The methodology section is hard to understand, bringing the pseudocode from Appendix to the main paper may help with this.
- The experiment section applies the proposed PGNN approach to various GNN backbone (SGC, GCN, SAGE), each compares to different baselines and on different benchmark datasets, which looks a bit confusing to me.
- The conclusion of each experiment is hard to find in the paper.

2. The performance improvements looks marginal.

**Questions:**

My major concerns are related to the clarity and the performance.

---

### Note · Authors · 2024-11-25

I have read and agree with the venue's withdrawal policy on behalf of myself and my co-authors.